# General Method to Increase Carboxylic Acid Content on Nanodiamonds

**DOI:** 10.3390/molecules27030736

**Published:** 2022-01-23

**Authors:** Ganesh Shenoy, Jessica Ettedgui, Chandrasekhar Mushti, Jennifer Hong, Kelly Lane, Burchelle Blackman, Hak-Sung Jung, Yasuharu Takagi, Yeonee Seol, Martin Brechbiel, Rolf E. Swenson, Keir C. Neuman

**Affiliations:** 1Laboratory of Single Molecule Biophysics, National Heart Lung and Blood Institute, National Institutes of Health, Building 50, Room 2134, Bethesda, MD 20892, USA; ganeshshenoy@outlook.com (G.S.); jennyh28@gmail.com (J.H.); haksung.jung@kriss.re.kr (H.-S.J.); takagiy@nhlbi.nih.gov (Y.T.); seoly@nhlbi.nih.gov (Y.S.); 2Chemistry and Synthesis Center, National Heart Lung and Blood Institute, National Institutes of Health, Building B, Room 3040, 9800 Medical Center Drive, Rockville, MD 20850, USA; jess.benjamini-ettedgui@nih.gov (J.E.); chandrasekhar.mushti@nih.gov (C.M.); kelly.lane@nih.gov (K.L.); burchelle.blackman@nih.gov (B.B.); 3Radiation Oncology Branch, National Cancer Institute, National Institutes of Health, Building 10, Room B3B69, Bethesda, MD 20892, USA; martinwb@mail.nih.gov

**Keywords:** fluorescent nanodiamond, optical trapping, TIRF, rhodium catalysis, functionalization

## Abstract

Carboxylic acid is a commonly utilized functional group for covalent surface conjugation of carbon nanoparticles that is typically generated by acid oxidation. However, acid oxidation generates additional oxygen containing groups, including epoxides, ketones, aldehydes, lactones, and alcohols. We present a method to specifically enrich the carboxylic acid content on fluorescent nanodiamond (FND) surfaces. Lithium aluminum hydride is used to reduce oxygen containing surface groups to alcohols. The alcohols are then converted to carboxylic acids through a rhodium (II) acetate catalyzed carbene insertion reaction with *tert*–butyl diazoacetate and subsequent ester cleavage with trifluoroacetic acid. This carboxylic acid enrichment process significantly enhanced nanodiamond homogeneity and improved the efficiency of functionalizing the FND surface. Biotin functionalized fluorescent nanodiamonds were demonstrated to be robust and stable single-molecule fluorescence and optical trapping probes.

## 1. Introduction

Nanodiamonds (NDs) were first produced by graphite detonation in the USSR in the 1960s [1]. Initially, NDs were used for their abrasive properties as lubricant additives [2] in industrial applications [1]. Subsequently, additional methods have been utilized to synthesize nanodiamonds, including chemical vapor deposition, laser ablation, and high-pressure–high-temperature (HPHT) approaches [3,4,5]. With the discovery of fluorescent properties arising from Nitrogen-Vacancy (NV) centers in their lattice structure, fluorescent nanodiamonds (FNDs) show great promise in many cutting-edge applications [3], including quantum computing, [6] quantum sensing [7], and nanophotonics [8,9]. FND research is particularly flourishing in biological and biomedical imaging, [10,11] drug-delivery, [12] and subcellular tracking [13,14,15] applications. FNDs are typically prepared by bombarding nitrogen doped HPHT diamonds with a high-energy (1–10 MeV, typically 2 MeV) electron beam, followed by annealing at elevated temperatures (typically 800–900 °C, 1–5 h, in vacuum or inert gas) to form NV centers by inducing vacancy diffusion. A final step consists of removing surface impurities and induced diamond graphitization by oxidation through treatment with nitric or sulfuric acid, or other oxidants [16,17].

FNDs exhibit good biocompatibility, low toxicity, and emission in the near-infrared, making them ideal for biomedical applications [18,19]. Their counterpart, organic fluorophores, often used in these applications, suffer from photobleaching and photoblinking, whereas FNDs afford indefinite photostability [20]. FNDs are particularly outstanding due to the chemical inertness of the diamond core, whereas the surface is highly tailorable, fully accessible, and contains a number of functional groups that can be used to adapt their surface properties to different environments [21].

Control of the ND surface chemistry is critical for reliable, efficient, and stable functionalization. A major drawback is the lack of uniformity in the chemical structure of the ND surface, which increases the likelihood of unintended side reactions and the loss of a controlled surface chemistry. Many methods are available to homogenize ND surfaces, such as hydrogenation and hydrogen annealing, to produce sp^2^ carbon, [22,23] direct oxidation to produce carboxylic groups, [24,25,26] ozone treatment followed by hydrolysis to generate carboxylic acids, [27,28] reduction of different oxygen containing groups to generate a hydroxylated surface, [29] and silylation [30].

Functionalization of FNDs with drugs, biomolecules, such as antibodies, as well as their incorporation into composites and hybrid materials for bio-medical applications [24] allow them to serve as targeted imaging probes [31]. This can be achieved via non-covalent or covalent interactions [5].

Many published carbon nanoparticle functionalization schemes utilize surface exposed carboxylic acids as “handles” onto which other groups are attached through well-established coupling chemistries [32,33]. Currently, carboxylic acids are generated on carbon nanoparticles primarily by treatment with nitric, sulfuric, or other acids [3,5]. However, these oxidative treatments do not selectively generate carboxylic acids but rather a mixture of oxygen containing groups, including ketones, aldehydes, epoxides, lactones, alcohols, and carboxylic acids [34,35,36,37]. This partitioning of the surface oxygen content among different functional groups reduces the yield of carboxylic acids and, thus, reduces the efficiency of subsequent surface coupling reactions.

With the goal of achieving simple and versatile strategies to functionalize FND particles, we have developed a method that enriches the surface density of carboxylic acids on oxidized detonation and HPHT fluorescent NDs. Two independent quantification methods using dynamic light scattering (DLS) and thermogravimetric analysis (TGA) verified increased carboxylic acid density on the surface of FNDs subjected to this carboxylic acid enrichment method. We implement and demonstrate the robustness of this synthetic method for FNDs functionalization with several commonly used tags in biological applications. Lastly, biotinylated FNDs synthesized with this method were used to calibrate the evanescent field in a total internal reflection fluorescence (TIRF) microscope by attaching them to an individual DNA molecule that was supercoiled via magnetic manipulation, and we demonstrate high-efficiency optical trapping of single biotinylated FNDs attached to individual DNA molecules.

## 2. Results and Discussion

### 2.1. Synthetic Method and Characterization

Different production methods generate NDs with different sizes, morphologies, and surface terminations. The diamond core is usually enclosed in a non sp^3^ carbon defective diamond shell, which is typically amorphous carbon and/or sp^2^ carbon [38,39,40]. The ND surface reactivity is strongly dependent on the nature of this outer shell, and therefore, needs to be carefully characterized so that subsequent wet chemistry can be adjusted accordingly. The three-step synthetic method presented in Figure 1 selectively enriches carboxylic acids on the surface of oxidized detonated nanodiamonds (NDs) and FNDs. First, a non-selective reducing agent, LiAlH_4_ (LAH), reduces the oxygen containing functional groups on the FND surface primarily into alcohols. Subsequently, rhodium (II) acetate catalyzes a carbenoid insertion reaction [41,42] to attach *tert*–butyl diazoacetate onto the alcohol groups through an ether bond. Lastly, the *tert*–butyl ester is cleaved with trifluoroacetic acid (TFA) to yield carboxylic acid. This procedure generates carboxylic acids linked to the nanoparticle surface by several atom spacers, which may reduce steric constraints and enhance the efficiency of subsequent surface chemistry.

The progressive modification of the ND surface functional groups was confirmed by Fourier-transform infrared spectroscopy (FTIR) (Figure 1) with detonation NDs. The initial surface of the NDs (Figure 1, red curve) comprised different oxygen-containing functional groups reflected in a carbonyl (C=O) stretch at ~1750 cm^−1^, a broad O-H stretch centered at ~3400 cm^−1^, a narrower O-H stretch at ~1630 cm^−1^, a sharp C-O-C asymmetric bending peak at ~1100 cm^−1^, and a peak at ~890 cm^−1^ that has been observed previously but not assigned [34]. In the first step, the oxygen containing groups on the detonation NDs surface were reduced into alcohols with LAH (Figure 1, blue curve). The carbonyl stretch at ~1750 cm^−1^ decreased, whereas the O-H stretch at 3400 cm^−1^ and the O-H bend at ~1630 cm^−1^ increased significantly, consistent with the conversion of carbonyl containing groups (such as ketones, aldehydes, lactones, and carboxylic acids) into hydroxyl groups [34]. In addition, the sharp C-O-C peak at ~1100 cm^−1^ decreased and a broad peak centered at ~1050 cm^−1^ appeared, consistent with the formation of primary and secondary alcohols [34]. As observed previously [34], the peak at ~890 cm^−1^ was lost in the reduction process. In the next step, an ester was formed via a rhodium catalyzed carbenoid insertion and the ester was cleaved with TFA leaving carboxylic acid coupled to the nanodiamond via an ether bond. The final IR spectrum (Figure 1, green curve) showed a decrease in the O-H stretch and an increase in the carbonyl peak at ~1750 cm^−1^. The weak peak at ~1440 cm^−1^ is consistent with a weak O-H bending peak associated with carboxylic acid. Furthermore, the appearance of a strong peak at ~1200 cm^−1^ confirms presence of an ether bond. The evolution of the FTIR spectrum through the carboxylic acid enhancement process provides evidence that the final carbonyl peak largely represents carboxylic acids rather than a mixture of oxygen-containing functional groups.

Once the surface of the NDs is enriched in carboxylic acids, it can be readily and systematically functionalized with other tags. First, treatment with thionyl chloride converts carboxylic acid into acyl chloride, then an amine–terminated tag can be coupled directly with the acid chloride of the NDs with the assistance of 4–dimethylaminopyridine (DMAP) as a catalyst and triethylamine (TEA) as base. The coupling efficiency of carboxylic acid enriched FNDs was verified with *O*–(2–aminoethyl)–O′–(2–azidoethyl) pentaethylene glycol (amine–PEG–azide), as depicted in Figure 2. Azides have a strong and unique infrared signature at ~2150 cm^−1^ [43], which facilitates verification of surface functionalization.

The azide stretching peak at ~2100 cm^−1^ is clearly visible in the spectrum of the functionalized diamonds that was not present in the carboxylic enriched diamonds prior to functionalization (Figure 2, red line). Additionally, the presence of amide I and amide II peaks [44] at 1650 cm^−1^ and 1570 cm^−1^, respectively, indicates covalent coupling of the amine–PEG–azide to carboxylic acid. Alkyl peaks at 2870 cm^−1^ and 2930 cm^−1^ are consistent with carbon–hydrogen bonds of the PEG linker. The broad peak centered at 3400 cm^−1^ is attributed to hydrogen-bonded N-H vibrations from the amide bond.

Using the same FND surface carboxylic acid enrichment method (Figure 1), we produced a series of FNDs covalently labeled with a heterobifunctional (PEG) linker containing an amine and one of several chemical groups commonly used for tagging biological molecules via click and other standard coupling chemistries (biotinylate, maleimide, active ester, amine, alkylating group, and azide), or directly with the protein tag systems HALOTAG, SNAP–TAG, and its derivative CLIP–TAG. Covalent attachment of each chemical group via amide bond formation was verified by FTIR and TGA (Appendix A). This efficient synthetic procedure was repeated systematically and successfully with the tags listed in Figure 3. Details of the synthesis and characterization are provided in the Appendix A.

### 2.2. Thermogravimetric Analysis (TGA) of FND Surface Functionalization

Molecules adsorbed to the surface of a nanoparticle, free molecules, physio-absorbed species, and loosely coordinated solvent molecules, can be discriminated by measuring the volatilization temperatures necessary to dissociate bound molecules from a nanoparticle surface [45,46]. TGA can quantify the surface density of chemical species absorbed on nanoparticles [45,47,48], such as carbon-based [49,50] (single and multiple wall nanotubes [51], fullerenes, and nanodiamonds [52]), metal nanoparticles (including gold, silver, iron, and copper nanoparticles), metal oxide nanoparticles (iron oxide, ZnO, TiO_2_, CeO_2_, SiO_2_, among others), and quantum dots where ligand coverage, exchange, and removal can be quantified [45,47,48,53].

Similarly, the total mass of organic compounds grafted onto FND core samples can be determined by the first derivative of a TG curve, termed Derivative Thermogravimetric Analysis (DTGA or DTG). Grafted adducts per ND can be calculated directly from the mass ratio after heating, [54] providing a direct measure of the carboxylic acid enrichment on the FND surface [52]. Figure 3 shows the TGA and DTG curves of FND–amido–PEG_6_–biotin with PEG_6_–biotin grafted on the ND surface, using the same synthetic route described earlier (Figure 2). The identities of the compounds in the ND sample were identified by FTIR.

The TGA curve of FND–amido–PEG_6_–biotin is presented in Figure 3A (top) along with its first derivative (bottom), which emphasizes the temperatures at which changes in mass occur. The DTG curve shows a complex pattern compared to the DTG curves of the untreated FNDs and acyl chloride FNDs (Appendix A). Each peak is detected in the DTG curve and reported on the TGA graph to quantify the mass loss over each temperature range. By 600 °C, FND–amido–PEG_6_–biotin lost ~3.7 wt %, which includes all volatiles, semi-volatiles, and non-volatile compounds. Free Amine–PEG_6_–biotin decomposes in this temperature range and starts to degrade rapidly at ~300 °C in argon. Two sub-steps are identified in the free Amine–PEG_6_–biotin DTG curve with a corresponding shoulder at 350.6 °C, the maximum weight loss temperature (T_max_) occurs at 397.4 °C (Figure 3B, solid line), and is lost completely above 515 °C. In comparison, FND–amido–PEG_6_–biotin exhibits the highest loss rate at ~465 °C, attributed to the amido–PEG_6_–biotin moiety covalently bound to the ND surface (Figure 3B, dashed line). Nonetheless, at ~400 °C a small peak is observed in the FND–amido–PEG_6_–biotin DTG curve, which is identified as non-covalent bound fraction of amine–PEG_6_–biotin. The degradation temperature of free PEG_6_–biotin is expected to occur at a lower temperature compared to covalently bonded PEG_6_–biotin to the FND surface because higher heat is necessary to volatilize covalently bonded molecules [55]. If we account for the mass loss of the acylated FNDs over the full temperature range then 0.42 wt % of the total mass of FND–amido–PEG_6_–biotin is lost during this step (green shaded region, Appendix A), which corresponds to ~4300 PEG_6_–biotin molecules on the FND surface (see calculations in Appendix A). By measuring the overall mass loss before and after functionalization and verifying that the free Amine–PEG_6_–biotin was completely volatized, leaving no residual carbon mass, we reduced artifacts associated with mass-loss of the bare NDs and with possible carbonization of the ligand. In the unlikely event that carbonization of the covalently coupled ligands occurred despite the complete loss of free Amine–PEG_6_–biotin above 515 °C, the density of attached ligands would be underestimated.

### 2.3. Streptavidin Biotin–FND Coupling

The carboxylic acid content available for functionalization on FND surfaces was monitored by taking advantage of the high affinity of streptavidin for biotin. With a dissociation constant (K_d_) on the order of ≈10^−14^ mol/L, the binding of biotin to streptavidin is one of the strongest non-covalent interactions [56]. Carboxylic acid enriched FNDs coupled with amine–PEG–biotin were titrated with free streptavidin. Due to the tetravalency of streptavidin, the biotin–PEG–FNDs formed streptavidin-mediated aggregates at low streptavidin concentrations and isolated streptavidin-coated biotin–PEG–FNDs above a critical streptavidin concentration, at which the surface biotins are saturated with streptavidin. This streptavidin-dependent aggregation and eventual surface saturation of the biotin–PEG–FNDs can be monitored through changes in the FND hydrodynamic diameter measured using dynamic light scattering (DLS). The surface density of available biotins can be inferred from the concentration of streptavidin that saturates the biotin– FNDs, resulting in a decrease in the hydrodynamic radius associated with the prevention of streptavidin-mediated aggregation. A sample of FNDs was separated into two batches, one was subjected to the carboxylic enrichment method outlined above and the other was used as purchased. Both batches were biotinylated using identical procedures and identical amounts of diamond, dichloromethane, DMAP, TEA, and amine–PEG_2000_–biotin. Conjugation with high molecular weight polyethylene glycol (PEG) linkers is necessary for samples used in DLS measurement to achieve colloidal stability of biotinylated FNDs. DLS measurements (Figure 4) and a fluorescence assay (Appendix A) were used to quantify the surface density of amine–PEG–biotin coupled to enriched and unenriched FNDs.

Figure 4 displays DLS measurements of the average hydrodynamic diameter of carboxylic acid enriched FND–biotin and unenriched FND–biotin as they are titrated with streptavidin. Without streptavidin, the nanodiamonds are dispersed and have an average particle size of 170 nm. Due to the tetravalence of streptavidin, biotinylated nanoparticles aggregate with the addition of sub-saturating amounts of streptavidin via crosslinking [57]. Enriched FND–biotin displayed a characteristic increase in hydrodynamic diameter (~220nm) with increasing concentrations of streptavidin. Once the streptavidin concentration is equal to the biotin concentration, the diamonds remain dispersed, leading to a decrease in the average particle size. The critical concentration, (i.e., at which all biotins on the FNDs surface are bound to streptavidin), is reached at a streptavidin concentration of ~30 nM, at which point the hydrodynamic diameter decreases to a value close to, but slightly larger than, the original size (180 nm). This critical concentration is a good approximation of the concentration of accessible biotin on the FNDs (Figure 4, red arrow).

These measurements indicate a surface density of ~600–700 biotins per FND (see calculations in Appendix A). The unenriched FND–biotin sample showed no change in average hydrodynamic diameter with increasing streptavidin concentration, indicating a negligible surface density of conjugated biotin. These results demonstrate that our carboxylic acid enrichment process significantly increased the functionalization of FNDs with biotin. The FNDs–biotin titration with streptavidin is an indirect way to evaluate the amount of carboxylic acids available for conjugation. Not all carboxylic acids present on the surface of the FNDs will be accessible for functionalization.

The comparison of biotin surface density calculations using TGA and DLS measurements is inconclusive. Due to technical limitations, each method evaluated a different PEG liker size; PEG_6_ for TGA and PEG_2000_ for DLS. Although high molecular mass PEG_2000_ is necessary to stabilize colloidal FNDs in PBS solution and generate reliable DLS measurements, its thermal decomposition is complex and, thus, the TGA measurement is not informative. TGA measurements shows a higher PEG_6_–biotin surface density which can be rationalized by a denser packing of PEG_6_ on the FNDs surface. The high molecular mass PEG_2000_ is expected to have a radius of gyration more than 10-fold larger than the low molecular mass PEG_6_. The increased steric crowding of the larger radius of gyration limits the amount of PEG biotin that can bind to the FNDs surface. Nonetheless, the enrichment process significantly increased the density of carboxylic acids available for functionalization.

### 2.4. Functionalized FNDs as Single-Molecule Probes

We demonstrated the applicability of biotinylated FNDs as single-molecule fluorescence and optical trapping probes. Total internal reflection fluorescent (TIRF) microscopy suppress background fluorescence by generating an evanescent wave through total internal reflection of the excitation laser at the glass water interface of the slide [58]. The evanescent wave intensity decays exponentially away from the surface (*z*-direction) with a characteristic length (the penetration depth, *d*) that is difficult to measure in situ. FNDs are ideal probes to characterize the intensity profile in *z* due to their exceptional optical stability [30]. ND–biotin particles were bound to streptavidin coated magnetic beads tethered by individual DNA molecules to the surface of a microscope slide. The magnetic beads were rotated via external magnets in the counter-clockwise direction to overwind (positively supercoil) the DNA, which results in the beads approaching the surface (decreasing *z*) as the DNA forms plectonemes (Figure 5A) [59]. The fluorescent intensity from the FNDs bound to the magnetic bead is directly correlated with the height change of the bead providing the intensity profile as a function of *z*-position, which can be fit to obtain the characteristic penetration depth, *d* (Figure 5A,B).

We also demonstrate the use of ND–biotin as an optical trapping probe to apply force on individual DNA molecules. Smaller particles with higher refractive indices improve the spatial and temporal resolution in optical trapping measurements [60]. As diamond has a higher refractive index (*n* = 2.4) than polystyrene (PS) (*n* = 1.57), ND particles have been shown to be superior to PS beads as optical trapping probes. For example, the trapping efficiency of silica encapsulated ND is 7-fold higher than that of the same sized PS bead [61]. Nonetheless, FNDs have not been adopted as force transducers in optical trapping experiments, due in part to difficulties in functionalization and attachment of nanodiamonds. FND–biotin particles labeled with streptavidin were tethered via the biotinylated end of DNA, the other end of which was attached to the surface of a slide (Figure 5C). DNA tethered FNDs were stably trapped, allowing us to estimate the trap stiffness by power spectrum analysis and also to apply tension on the DNA molecules (Figure 5C,D). The average stiffness measured from 10 ND–biotin particles (100 nm nominal diameter) was 0.85 ± 0.06 fN/nm (mean ± standard error of the mean) at 20 mW of trapping laser power. Unexpectedly, the optically trapped FNDs emitted red fluorescence (Figure 5C and Appendix A) presumably due to CW two-photon 1064 nm excitation of NV^-^ centers by the optical trapping laser [20,62,63]. This finding opens the possibility of two-photon based position measurements of optically trapped particles that have previously been limited by photobleaching of conventional fluorophores [64].

## 3. Materials and Methods

### 3.1. Materials

Detonated nanodiamonds and fluorescent detonated nanodiamonds (80 nm diameter) were supplied and reduced by Adámas Nanotechnologies, Inc., Raleigh, NC, USA. Reduced HPHT fluorescent nanodiamond were produced by static synthesis and annealed by Adámas Nanotechnologies. We used thionyl chloride (97%, Sigma-Aldrich, Saint Louis, MO, USA), methanol 99.8% anhydrous (Sigma-Aldrich), tetrahydrofuran (THF) 99.9% Extra Dry (Sigma-Aldrich), dichloromethane (DCM) 99.8% anhydrous (Sigma-Aldrich), and N,N–dimethylformamide (DMF) 99.8% anhydrous (Sigma-Aldrich), 99% trifluoroacetic acid (Sigma-Aldrich). All reagents were used without additional purification.

DLS particle size measurements were made with a Malvern Zetasizer Nano ZS (Malvern, Malvern, UK) equipped with a 633 nm laser. Particle size distributions are the average of 100 30 s scans and zeta potential of 300 scans. The bulk refractive index of bulk diamond (2.4) was used.

TGA measurements were performed with ~10 mg of material in a platinum pan using a Q50 thermogravimetric analyzer (TA Instruments, New Castle, DE, USA). All measurements were performed in an Ar atmosphere.

Elemental analysis (Carbon, Hydrogen, and Nitrogen) was performed by Robertson Microlit Laboratories with ~1.5 mg of material in a Perkin-Elmer Model 2400 CHN Analyzer (Perkin-Elmer, Waltham, MA, USA) with helium carrier gas, oxygen combustion gas, and an operating temperature of ~950 °C. The instrument was calibrated with NIST141D acetanilide.

NMR analysis was attempted with functionalized nanodiamonds in aqueous suspension at various concentrations using a Bruker 400 MHz spectrometer (Bruker, Billerica, MA, USA). Neither 1D nor 2D (DOSY) measurements revealed chemical shifts associated with the functionalized nanodiamonds. We attribute the lack of signal to the inability to prepare solutions with sufficient concentration of surface ligands given issues with nanodiamond aggregation at high concentrations and the large size of the nanodiamonds, which both decrease the relative concentration of surface ligands and reduce the tumbling time, resulting in signal degradation [65].Furthermore, solid phase NMR of the functionalized nanodiamonds is impractical due to the loss of sensitivity arising from the paramagnetic centers in the nanodiamonds and the relatively low density of functional groups on the large (~100 nm) nanodiamonds [66].

### 3.2. General Method for Reduction of FNDs

Detonated NDs and FND powder were reduced with 2 M lithium aluminum hydride in THF solution for 48 h at 65 °C. The dispersion was cooled in an ice bath and quenched slowly with drop-wise addition of 1 M HCl. The NDs were then thoroughly washed with 1 M HCl, 1 M NaOH, and, finally, DI water until neutral pH was reached. The reduced and purified NDs were subsequently lyophilized into a powder form and used as a starting material for subsequent functionalization.

### 3.3. General Method for Carboxylic Acid Enrichment on Reduced FNDs

Reduced detonation and reduced HPHT fluorescent NDs (70 mg) were dried under vacuum by heating with a heat gun for 10 min and cooled under an argon environment. NDs were dispersed with sonication in anhydrous benzene (1.5 mL) with rhodium (II) acetate dimer (1.5 mg, Sigma-Aldrich) and *tert*–butyl diazoacetate (20 µL, Sigma-Aldrich). The vial was flushed with argon, sealed, and kept at 80 °C for 48 h. The NDs were subsequently isolated by centrifugation and washed thoroughly with dichloromethane (DCM) to remove any unreacted reagents and dried for several hours under vacuum. Trifluoroacetic acid (2 mL) was added to the dried NDs and dispersed by sonication. After 48 h at room temperature, the NDs were isolated by centrifugation and washed thoroughly with water and methanol and dried under vacuum. A small amount of the NDs was used to obtain an FTIR spectrum.

### 3.4. Coupling Carboxylic Acid Enriched FNDs with Tags

Enriched FNDs (30 mg) were dried under vacuum with a heat gun for 10 min. Neat thionyl chloride (5 mL) was added to the ND containing sealed vial, followed by sonication to suspend the NDs. The dispersion was stirred and heated at 70 °C for 48 h. The thionyl chloride supernatant was removed after cooling the vial to room temperature and centrifugation (3270× *g* for 1 h). The pellet of acyl chloride functionalized FNDs was kept under a vacuum for 4 h to remove residual thionyl chloride. Dry chloroacetylated FNDs (25 mg) were suspended in dry DCM (2 mL) with amino-tags, DMAP (50 mg) and triethylamine (100 µL). The FND suspension was kept at 39 °C for 24 h. The functionalized FNDs were isolated by centrifugation, thoroughly washed with DCM and methanol to remove unreacted reagents and dried under vacuum. A few milligrams of the dried powder were used to obtain an infrared spectrum and TGA. (See Appendix A).

### 3.5. Optical Trap

The optical trap is built around an Olympus IX-71 inverted microscope with a UPlanApo/IR 60× magnification 1.20 numerical aperture water immersion objective (Olympus, Center Valley, PA, USA) using a 5W 1064 nm wavelength laser (J20I-BL-106C; Spectra-Physics, Mountain View, CA, USA) as the trapping laser source. Fluorescence images were acquired using an Andor iXon DV897EC-BV EMCCD camera (Andor-Oxford Instruments, South Windsor, CT, USA) in combination with a TRITC long-pass widefield imaging filter cube (Olympus, Center Valley, PA, USA). Typically, a laser power of ~20 mW was used to trap the particles. A power spectrum was collected for each trapped FND and the roll-off frequency, *f*_0_, was determined by fitting the power spectra with a Lorentzian function [60]. The trapping stiffness, *k*, is related to the roll-off frequency: $f_{0}=k{\left(2\pi \beta \right)}^{-1}$, where β=6πηa is the hydrodynamic drag on a particle of radius *a*, and η is the viscosity of water. We use the nominal size of the nanodiamonds (100 nm diameter) to compute the average optical trap stiffness. Although we cannot determine the size of each optically trapped diamond in situ, we note that the roll-off frequency is expected to be proportional to the square of the particle size. Comparing the average roll-off frequency (143 Hz) to the standard deviation (29 Hz) allows us to estimate a size dispersion (standard deviation divided by the mean) of ~10% among the 10 FND samples measured.

## 4. Conclusions

The use of FNDs in various applications relies on the controlled and efficient surface functionalization. However, the surface chemistry is not always well controlled, but rather partitioned among many different groups, decreasing the efficiency, and increasing the possibility for unintended side reactions while pursuing further surface functionalization.

We have improved current methods with a process to effectively convert the large mixture of oxygen containing groups, obtained through the commonly employed oxidative treatments of FNDs, primarily into carboxylic acids. Our method consists of an initial reduction using a non-selective reducing agent, such as lithium aluminum hydride, to convert the oxygen content on the FNDs surface into alcohols. The alcohols are subsequently converted selectively into carboxylic acids through a rhodium catalyzed carbenoid insertion with *tert*–butyl diazoacetate and subsequent ester cleavage with trifluoroacetic acid. We showed that the nanodiamond surface was enriched in carboxylic acid by measuring the relative amount of amine–PEG–biotin coupled onto carboxylic acid enhanced and control fluorescent nanodiamonds.

Improving the amount of carboxylic acid on the nanoparticle surface increases the efficiency of subsequent conjugation reactions. We have shown the robustness of our method and applied this synthetic procedure to a wide selection of biologically relevant tags. The tags were systematically and successfully attached to the FND surface demonstrating the broad applicability of our approach. This process could be applied to a larger variety of carbon-based nanoparticles, including carbon nanotubes and graphene structures. Highly oxidizing environments, either through acid treatment or high temperature in an oxygenated environment, are often used with carbon based nanoparticles to generate a broad mixture of oxygen-containing functionalities including: ketones, aldehydes, alcohols, epoxides, lactones, and carboxylic acids [21,22,23,49]. Our method homogenizes the nanoparticle surface and increases the overall carboxylic acid content, which is essential for effective carboxylic acid-based surface conjugation of many different biological molecules.

In addition, the functionalization was robust in physiological conditions and stable, as demonstrated by the application of FNDs to further applications including fluorescence imaging and optical trapping.

## 5. Patents

The work reported in this manuscript resulted in published patent Method for Functionalizing Carbon Nanoparticles and Compositions. US10,995,004 (5 May 2021).

## Data Availability

Not applicable.

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
