# Peer review of "General Method to Increase Carboxylic Acid Content on Nanodiamonds"

_molecules, 2022, doi:10.3390/molecules27030736_

Round 1

Reviewer 1 Report

The paper describes the way to functionalize the surface of fluorescent nanodiamonds, which should be of a great interest to material scientists as well as those working in biophysics and cell biology. The successful covalent labeling with a variety of chemical groups and tagging biological molecules seems the method in the manuscript promising in expanding the application of FNDs in biology.

I have the following minor comments.

1) "TRIF " in Keywords should be "TIRF".

2) page 6, L185

..and acid chloride FNDs (SI).

Should this be "acryl chloride FNDs"?

3) The stiffness of optical trapping also depends on the size and the shape of the material. The size of FND has a certain distribution. Their shape varies significantly. I am wondering how the authors took the control of these uncontrollable variables when they determined the stiffness with relatively small variation as 0.06 fN nm-1.

4) The resolution of the Figure 5 image needs to be improved. Labels are not readable.

5) Figure 5c. The fluorescence image of a trapped FND is hard to recognize. It could be reasonable if this is a single fluorescence image, but not an averaged image of multiple frames. It is helpful to describe the exposure time of iXon in the method for better understanding of this result.

Author Response

The paper describes the way to functionalize the surface of fluorescent nanodiamonds, which should be of a great interest to material scientists as well as those working in biophysics and cell biology. The successful covalent labeling with a variety of chemical groups and tagging biological molecules seems the method in the manuscript promising in expanding the application of FNDs in biology.

I have the following minor comments.

1) "TRIF " in Keywords should be "TIRF".

Corrected

2) page 6, L185

..and acid chloride FNDs (SI).

Should this be "acryl chloride FNDs"?

Corrected

3) The stiffness of optical trapping also depends on the size and the shape of the material. The size of FND has a certain distribution. Their shape varies significantly. I am wondering how the authors took the control of these uncontrollable variables when they determined the stiffness with relatively small variation as 0.06 fN nm-1.

The reviewer raises a good point concerning the variability of the diamonds and the resulting variability in the optical trap stiffness measurements.  The reported stiffness is the average stiffness measured for 10 individual nanodiamonds and the variation represents the standard error of the mean of the 10 measurements.  As the reviewer correctly points out the shape and size of the FNDs is variable and difficult if not impossible to determine on a particle by particle basis in the optical trap.  For this reason we report the average stiffness of 10 FNDs, reasoning that this will represent the average behavior of the nominally 100 nm FNDs.  To address this point in more detail, we note that the roll-off frequency of the particles in the optical trap is expected to vary in proportion to the square of the particle radius.  The relatively low standard deviation (29 Hz) of the roll-off frequency in comparison with the mean (143 Hz) indicates that the variability (standard deviation) in the size of the trapped diamonds is on the order of 10%. We have clarified these uncertainties and the approach we took to obtain a representative average value and uncertainty in the revised manuscript.

4) The resolution of the Figure 5 image needs to be improved. Labels are not readable.

We apologize for the poor resolution of the figure. We have replaced all the figures in the revised manuscript with high resolution images to improve the clarity and readability.

5) Figure 5c. The fluorescence image of a trapped FND is hard to recognize. It could be reasonable if this is a single fluorescence image, but not an averaged image of multiple frames. It is helpful to describe the exposure time of iXon in the method for better understanding of this result.

The reviewer has raised a good point concerning the contrast and visibility of the fluorescent nanodiamond in the image.  This image is a single frame (40 ms) from a short movie in which the trapped tethered FND is oscillated by moving the stage back and forth.  We have included the movie as a supplemental file, and provided the exposure time of the iXon camera to clarify the admittedly poor contrast in the single frame. The motion of the optically trapped FND is clear in the short movie.

Reviewer 2 Report

Shenoy et al. present in this manuscript the development of a method to specifically enrich the carboxylic acid content on fluorescent nanodiamond surfaces, as well as the functionalization of this surfaces by linking surface carboxyl groups to various organic fragments. The aim of the work is clear and the manuscript is well-written, therefore, results deserve publication. It is, however, surprising that the structural identification of the synthesized systems is weak. This latter is rather based on chemical knowledge than solid spectroscopic evidences. Both the applied IR and thermal analytical methods are regarded as supplementary qualitative techniques. I miss the application of elemental analysis and various NMR techniques for characterizing especially the functionalized systems, as well as Raman spectroscopy for characterizing the nanodiamond core. IR and TG cannot provide information about the level of functionalization; note that the IR intensity of various functional groups are very different and varied in derivatives, and thermolysis leads not only to the elimination of functional groups from the nanodiamond surface but carbonization as well.

Further comments:

--- IR spectra are only partially assigned. E.g. there are relatively strong IR bands in the spectrum of ND at about 1100 and 800 cm–1, which are missing in IR spectrum of reduced ND after COOH enrichment (Figure 1). Please explain.

--- All comments relevant to an IR spectrum requires presenting the spectrum at least in the supporting material, e.g. “The appearance of a strong peak at ~1200 cm-1 confirms the formation of an ether bond of the tert-butyl ester”; lines 113 and 114. The spectrum is also important to see IR bands in the C-H stretching region.

--- All FNDs functionalized with chemical linkers require proper characterization; see e.g. Scheme 3.

--- Calculation of the number of attached functional groups, e.g. PEG6-biotin molecules, on the nanodiamond surface based on mass loss in TG is questionable as thermal decomposition of organic molecules produce also carbon which remains on the surface; e.g. lines 182-202. Authors should comment on this. 

Author Response

Shenoy et al. present in this manuscript the development of a method to specifically enrich the carboxylic acid content on fluorescent nanodiamond surfaces, as well as the functionalization of this surfaces by linking surface carboxyl groups to various organic fragments. The aim of the work is clear and the manuscript is well-written, therefore, results deserve publication. It is, however, surprising that the structural identification of the synthesized systems is weak. This latter is rather based on chemical knowledge than solid spectroscopic evidences. Both the applied IR and thermal analytical methods are regarded as supplementary qualitative techniques. I miss the application of elemental analysis and various NMR techniques for characterizing especially the functionalized systems, as well as Raman spectroscopy for characterizing the nanodiamond core.

We appreciate the overall enthusiasm for the work from the reviewer.  To address the concerns of the reviewer related to the elemental analysis, we have included details of the elemental analysis in the revised supplementary information. The elemental analysis clearly reveals an increase in hydrogen and nitrogen associated with functionalization of the FNDs. Unfortunately, due to technical issues the company that performed the elemental analysis was unable to provide elemental analysis for the Azide, ND-amido-PEG3-amido-t-Boc-N, and SNAPtag samples.

We agree with the reviewer that NMR analysis of the functionalized NDs would provide additional verification of the functionalization specificity and provide quantitative and qualitative results concerning bound versus unbound ligands. Unfortunately, despite significant efforts to characterize the ligands on the FNDs using standard NMR techniques in addition to more sophisticated 2-D techniques such as DOSY, we were unable to achieve sufficient resolution. Since we were unable to characterize the functionalization of the NDs by NMR we characterized the functionalization of the NDs through other approaches including IR, TGA, and functional assays.

IR and TG cannot provide information about the level of functionalization; note that the IR intensity of various functional groups are very different and varied in derivatives, and thermolysis leads not only to the elimination of functional groups from the nanodiamond surface but carbonization as well.

The reviewer is correct that IR cannot provide information concerning the level of functionalization due to the different absorption strengths of different groups.  The IR curves in Figures 1 and 2 demonstrate the changes in the chemical composition of the NDs consistent with the proposed schemes, but we rely on TGA and functional assays to estimate the levels of functionalization achieved.

The reviewer is correct that TG will lead to carbonization as well as elimination of the functional groups from the NDs.  We account for carbonization by running a control on the starting materials (ligands and  bare nanodiamonds) before functionalization (line 214). This control reveals the degree of carbonization of the ligands, which was minimal for the ligands tested.

Further comments:

--- IR spectra are only partially assigned. E.g. there are relatively strong IR bands in the spectrum of ND at about 1100 and 800 cm–1, which are missing in IR spectrum of reduced ND after COOH enrichment (Figure 1). Please explain.

The reviewer raises a good point.  We were focused on the large and expected changes in the IR spectrum of the NDs during the reduction and functionalization process that are below the difficult to assign fingerprint region (800-1500 cm‑1). Nonetheless, in the revised manuscript we have provided the identity of the other strong IR bands based on previous reports, with the exception of the strong absorption peak near 890 cm-1 in the oxidized ND sample (Figure 1, red curve), which has been previously observed, but not assigned.  Importantly, this peak is absent in the IR spectra of the reduced and functionalized NDs, consistent with previous reports indicating the loss of this peak in reduced NDs (reference 34).    

Briefly, the absorption peak at 1100 cm-1 corresponds to strong C-O and C-O-C signals associated with hydroxyl, ether, ester, lactone, and acid anhydride.

The broadening of the strong peak at 1100 cm-1 over the range of 1040-1060 and appearance of a weak peak at 1150 in the reduced sample are consistent with the formation of primary and secondary alcohols as expected from reduction of the surface groups.

We attribute the sharp apparent peak near 800 cm-1 in the reduced sample to background from absorbed water that also contributes to the strong absorption in the 3200-3600 cm-1 region.  This is a well-known background artifact that affects reduced diamond samples that retain absorbed water.

--- All comments relevant to an IR spectrum requires presenting the spectrum at least in the supporting material, e.g. “The appearance of a strong peak at ~1200 cm-1 confirms the formation of an ether bond of the tert-butyl ester”; lines 113 and 114. The spectrum is also important to see IR bands in the C-H stretching region.

We have indicated the strong ~1200 cm-1 peak on the final IR spectrum in the revised Figure 1, and have assigned the significant peaks in all of the spectra as suggested in the previous point.  

--- All FNDs functionalized with chemical linkers require proper characterization; see e.g. Scheme 3.

We have provided Elemental analysis of the functionalized NDs is included in the revised supplementary information.

--- Calculation of the number of attached functional groups, e.g. PEG6-biotin molecules, on the nanodiamond surface based on mass loss in TG is questionable as thermal decomposition of organic molecules produce also carbon which remains on the surface; e.g. lines 182-202. Authors should comment on this. 

The review raises a good point that we implicitly addressed by performing controls with the NDs and ligands separately.  However, the reviewer is correct that the potential artifacts arising from carbonization should be addressed explicitly, which we have done in the revised manuscript.

We have addressed this concern by demonstrating that the ligands alone are completely lost during the TGA heating process.  Nonetheless, it remains formally possible that there is some carbonization of the covalently attached ligands, which would result in a slight under estimate of the degree of functionalization of the NDs by the ligand.  We make both of these points explicitly in the revised manuscript and we thank the referee for pointing out this potential artifact.

Round 2

Reviewer 2 Report

I appreciate that the authors have corrected the manuscript and taken into account the reviewer's comments. Further correction of the manuscript, however, is still required. Specifically:

--- C,H,N Elem. Anal. Results are now included in the Supporting Information, however, it is not clear how these results were obtained. Instrument specifications and measurement conditions are missing. The analysis of these results are also missing in the manuscript (how these results are correlated with expected compound structures?).  

--- Figure 1: there is no “blue” area, which is mentioned in the caption.  

--- Requested IR spectra are still not provided in SI.

--- Authors inability to record NMR spectra has to be documented in the manuscript (in Experimental section); details of experimental conditions and spectrometer have to be provided. A comment on the reason for lack of success should also be provided. If these experiments were done, the result, or failure, is educational for readers.

--- “The carboxylic acid content available for functionalization on FND surfaces was monitored by taking advantage of the high affinity of streptavidin for biotin.” Conclusions in this chapter (2.3) do not seem to be logical, and not justified by experimental data (see Figure 4). Authors commented that “Without streptavidin the nanodiamonds are dispersed and have an average particle size of 170 nm”; this is OK. It is, however, conflicting, that: (1) in the case of unenriched FNDs, the particle size does not change upon the addition of sub-saturating amounts of streptavidin via cross-linking; only increases at about 35 nM streptavidin concentration and then falls back. Unenriched FNDs, in principle, require much less streptavidin than enriched FNDs to reach saturation. (2) Why the 30 nM streptavidin concentration was selected as the critical concentration? The particle size increases at 40 nM and falls back at 45 mM. Altogether, the method seems to be very unreliable. 

Major correction
